# Intramuscular IL-10 Administration Enhances the Activity of Myogenic Precursor Cells and Improves Motor Function in ALS Mouse Model

**DOI:** 10.3390/cells12071016

**Published:** 2023-03-26

**Authors:** Paola Fabbrizio, Cassandra Margotta, Jessica D’Agostino, Giuseppe Suanno, Lorenzo Quetti, Caterina Bendotti, Giovanni Nardo

**Affiliations:** Laboratory of Molecular Neurobiology, Department of Neuroscience, Istituto di Ricerche Farmacologiche Mario Negri IRCCS, Via Mario Negri 2, 20156 Milan, Italy

**Keywords:** Amyotrophic Lateral Sclerosis, mouse models, skeletal muscle, macrophages, myogenic precursor cells

## Abstract

Amyotrophic Lateral Sclerosis (ALS) is the most common adult motor neuron disease, with a poor prognosis, a highly unmet therapeutic need, and a burden on health care costs. Hitherto, strategies aimed at protecting motor neurons have missed or modestly delayed ALS due to a failure in countering the irreversible muscular atrophy. We recently provided direct evidence underlying the pivotal role of macrophages in preserving skeletal muscle mass. Based on these results, we explored whether the modulation of macrophage muscle response and the enhancement of satellite cell differentiation could effectively promote the generation of new myofibers and counteract muscle dysfunction in ALS mice. For this purpose, disease progression and the survival of SOD1G93A mice were evaluated following IL-10 injections in the hindlimb skeletal muscles. Thereafter, we used ex vivo methodologies and in vitro approaches on primary cells to assess the effect of the treatment on the main pathological signatures. We found that IL-10 improved the motor performance of ALS mice by enhancing satellite cells and the muscle pro-regenerative activity of macrophages. This resulted in delayed muscle atrophy and motor neuron loss. Our findings provide the basis for a suitable adjunct multisystem therapeutic approach that pinpoints a primary role of muscle pathology in ALS.

## 1. Introduction

Amyotrophic lateral sclerosis (ALS) is the most common form of motor neuron (MN) disease [1]. It is characterized by the progressive degeneration of upper and lower motor neurons (MNs), leading to muscle atrophy, weakness, and spasticity [2]. Respiratory muscle denervation is typically the cause of death, and it usually occurs within five years of the onset of the disease. Of ALS cases, 90% are sporadic, while 10% are inherited [2] and are most frequently associated with mutations in SOD1, TARDBP, FUS, and C9ORF72 gene sequences.

The clinical similarities between the sporadic and familial forms of ALS imply that understanding the mechanisms responsible for familial ALS could shed light on both types of the disease [2].

The notion that MN death leads to ALS is commonly acknowledged. Nonetheless, some ambiguity remains about the sequence in which MN degeneration occurs in ALS. Several investigations in mouse models have demonstrated malfunction or decay of the neuromuscular junction preceding MN loss [3,4]. Additionally, distal axonopathy has been observed to occur before neuronal degeneration and disease onset [5,6,7]. Muscle atrophy precedes this sequence of events [8,9]. Indeed, the selective expression of mutant SOD1 in the skeletal muscles of mice is sufficient to induce ALS neurodegeneration, suggesting an intrinsic muscle pathology directly impacting MNs [10,11].

The inflammatory response differs in its contribution in the peripheral and central nervous system. While the activation of aberrant glial cells, infiltration of T cells, and release of proinflammatory molecules result in neurodegeneration in the CNS, the coordinated efforts of immune cells, including the removal of cellular debris and the release of wound-healing factors, are crucial for successful axon and muscle regeneration [12,13]. This may explain the association between higher peripheral inflammation and slower disease progression in mutant SOD1 ALS patients [14].

We recently showed that macrophage (MΦ) muscle infiltration is pivotal in driving the generation of new myofibers, lessening denervation atrophy and improving motor ability in ALS mouse models [15].

Muscle regeneration depends on satellite cells (SCs) placed on the muscle fiber surface [16]. Upon muscle damage, SCs are triggered to exit quiescence and undergo proliferation. Postmitotic myogenic precursor cells resulting from SCs then form multinucleated myotubes, leading to terminal myofiber differentiation and growth. During this process, MΦs are the most abundant immune cell recruited within the damaged tissue where they directly interplay with SCs and orchestrate their fate through different cues [17,18,19]. Among them, IL-10 is a potent immunomodulatory factor able to autoregulate the inflammatory response [20]. Indeed, IL-10 promotes the M1 to M2 shift of MΦs, which is a mandatory step to gaining steady muscle growth and regeneration [17]. Furthermore, IL-10 directly affects SC differentiation, escorting the transition of myogenesis from the proliferative to the differentiation stage in the injured muscle [21].

In this study, we evaluated the possibility of modulating the MΦ fingerprint and sustaining SC conversion to mature myofibers within the skeletal muscle of ALS mouse models. We showed that intramuscular IL-10 injections in transgenic SOD1G93A mice counteract skeletal muscle atrophy, directly stimulating SC differentiation and MΦ polarization. This eventually led to MN preservation and improved the motor performance of transgenic mice.

## 2. Materials and Methods

### 2.1. Animals

Transgenic SOD1G93A male and female mice with C57BL/6J (C57-SOD1G93A) and 129S2/Sv (129Sv-SOD1G93A) genetic backgrounds and their corresponding non-transgenic (Ntg) littermates were used in this study [22,23]. The animals were housed under SPF (specific-pathogen-free) standard conditions (22 ± 1 °C, 55 ± 10% relative humidity, and a 12 h light/dark schedule). There were 3–4 mice per cage, and the animals had free access to food (standard pellets, Altromin, MT, Rieper) and water.

The tibialis anterior (TA), gastrocnemius medialis (GCM) and quadriceps (QC) were treated daily with recombinant mouse IL-10 (PeproTech, London, UK) at a dose of 0.25 μg/day (10 μL/muscle) as previously described [24]. Procedures involving animals and their care were conducted in conformity with the institutional guidelines of the Mario Negri Institute for Pharmacological Research (IRFMN), Milan, Italy. The IRFMN adheres to the principles set out in the following laws, regulations, and policies governing the care and use of laboratory animals: Italian Governing Law (D.lgs 26/2014; Authorisation n.19/2008-A, issued 6 March 2008 by the Ministry of Health); Mario Negri Institutional Regulations and Policies providing internal authorization for persons conducting animal experiments (Quality Management System Certificate– UNI EN ISO 9001:2015, Reg. N° 6121); the NIH Guide for the Care and Use of Laboratory Animals (2011 edition) and EU directives and guidelines (EEC Council Directive 2010/63/UE).

### 2.2. Behavioral Analysis

The longitudinal bodyweight loss was determined by dividing the weight of a single mouse at various time points by its maximum weight achieved. For the paw grip strength test, the mouse was placed on the wire lid of a conventional housing cage and gently pulled by the tail until it grasped the grid with all four paws. The lid was then gently turned upside down, and the latency time for the mouse to fall on the table was recorded up to a maximum of 90 s. Each mouse received up to three attempts, and the longest latency was recorded. The onset of motor deficit was considered when the mice showed the first signs of impairment (a latency less than 90 s) in the paw grip strength test. The disease end stage was defined as the point at which the animals could not right themselves within 10 s after being placed on each side. This time point was considered the measure of survival length.

### 2.3. Dissection of Skeletal Muscles 

TA, GCM, and QC muscles were collected from C57-SOD1G93A and 129Sv-SOD1G93A male mice and their relative Ntg littermates at 14 and 16 weeks of age, corresponding to the symptom onset (OS) and symptomatic stages based on the performance on the grip strength test. All mice and their relative Ntg littermates were intracardially perfused with 0.1 M PBS. Muscles from each mouse were dissected, snap-frozen in isopentane, cooled in liquid nitrogen, and stored at −80 °C until use. After weighing the TA, GCM, and QC muscles, only the TA muscles were used for immunohistochemical, biochemical, and biomolecular analyses.

### 2.4. Immunohistochemistry

The tissue dissection process involved freezing the TA with cooled isopentane and leaving the spinal cord in 4% PFA overnight at 4 °C. It was then rinsed and stored for 24 h in 30% sucrose and 0.01 M PBS. For the muscles, 20 μm longitudinal and 10 μm transversal serial TA cryo-sections were collected on poly-lysine objective slides (VWR International, Milan, Italy).

To determine the cross-sectional area (CSA) and fiber type, the sections were incubated with MyHC type I (BA-D5, 1:10; DSHB, Iowa City, IA, USA), MyHC type IIa (SC-71,1:17; DSHB, Iowa City, IA, USA), MyHC type IIb (BF-F3, 1:9; DSHB, Iowa City, IA, USA), Rabbit anti-Laminin (1:100, L9393; Sigma-Aldrich, St. Louis, MO, USA), primary antibodies and respective secondary antibodies, anti-MIgG2b Alexa-flour 564 (A21144,1:500) (Invitrogen, Waltham, MA, USA), anti-MIgG1 Alexa-flour 488 (A21121, 1:500) (Invitrogen, Waltham, MA, USA), anti-MIgM Alexa-flour 647 (A21046,1:500; Invitrogen, Waltham, MA, USA), and anti-Rabbit Alexa Fluor 405 (ab175649, 1:500; Abcam, Cambridge, UK).

To assess the polarization of macrophages, longitudinal muscle TA sections were treated with acetone for 10 min, air-dried, and then washed. The muscle slides were blocked with 10% NGS or NDS in PBS for 1 h and then incubated overnight with primary antibodies: anti-CD11b, rat (1:200; Bio-Rad, Hercules, CA, USA); anti-iNOS, rabbit (1:200; Invitrogen); anti-mannose receptor, rabbit (1:500; Abcam, Cambridge, UK) at 4 °C. Secondary antibodies were as follows: Alexa488, anti-rat and Alexa647, anti-rabbit (1:500; Thermo Fisher, Waltham, MA, USA). The nuclei were counterstained with Hoechst (1:1000; Roche, Basel, Switzerland).

The spinal cord was cut into 30 µm serial transverse sections. The following primary antibodies and staining were used: anti-GFAP, mouse (1:2500; Merck, Darmstadt, Germany); anti-Iba1, rabbit (1:200; Fujifilm Wako Chemicals, Richmond, VA, USA); anti-ChAT, goat (1:200; Sigma-Aldrich, St. Louis, MO, USA). Secondary antibodies used were Alexa488 anti-mouse, Alexa647 anti-rabbit (1:500; Thermo Fisher, Waltham, MA, USA), and Alexa647 anti-goat (1:500; Thermo Fisher, Waltham, MA, USA).

The cells were treated with 4% PFA in PBS for 15 min to fix them, followed by permeabilization with 0.1% Triton for 5 min. To prevent unspecific signals, the cells were blocked with 1% BSA for 30 min. The primary antibodies anti-MF20, mouse (1:50; DSHB, Iowa City, IA, USA); anti-CD11b, rat (1:200; Bio-Rad, Hercules, CA, USA); anti-mannose receptor, rabbit (1:500; Abcam, Cambridge, UK); and anti-Ki67, rabbit (1:200; Abcam, Cambridge, UK) were diluted in blocking solution and incubated overnight at 4 °C. The cells were incubated with the secondary antibodies Alexa488 anti-rabbit, Alexa488 anti-mouse, and Alexa647 anti-rat (1:500; Thermo Fisher, Waltham, MA, USA) for 1 h at room temperature and washed with PBS. Nuclei were counterstained with Hoechst (1:1000; Roche, Basel, Switzerland) in PBS, and glasses were mounted in Fluorsave Mountant (Merck, Darmstadt, Germany). Images were captured using an A1 Nikon confocal microscope at 20× or 40× magnification in sequential scanning mode, and NIS Elements software 4.5 (Nikon, Melville, NY, USA) was used for data acquisition.

All the immunofluorescences were verified with control experiments using only the secondary antibody.

### 2.5. Real-Time PCR

RNA was extracted from the TA muscles using Trizol (Invitrogen, Waltham, MA, USA) and purified with PureLink RNA columns (Life Technologies, Carlsbad, CA, USA). The RNA samples were then treated with DNase I and reverse-transcribed using the High-Capacity cDNA Reverse Transcription Kit (Life Technologies, Carlsbad, CA, USA). A real-time PCR was carried out on the cDNA specimens in triplicate, using the Taq Man Gene expression assay (Applied Biosystems, Waltham, MA, USA) according to the manufacturer’s instructions, with 1x Universal PCR master mix (Life Technologies, Carlsbad, CA, USA) and 1x mix containing specific receptor probes (Life Technologies, Carlsbad, CA, USA). Relative quantification was calculated by determining the ratio between the cycle number (Ct) at which the signal crossed a threshold set within the logarithmic phase of the target gene and that of the reference β-actin gene (4310881E; Life Technologies, Carlsbad, CA, USA). The mean values of the triplicate results for each animal were used as individual data for a 2-ΔCt statistical analysis. The probes used for the real-time PCR were tumor necrosis factor-alpha (TNF-alpha) (Mm00443258_m1; Life Technologies, Carlsbad, CA, USA), interleukin 1β (Il-1β; Mm00434228_m1; Life Technologies, Carlsbad, CA, USA), and interleukin 10 (Il-10; Mm00439614_m1; Life Technologies, Carlsbad, CA, USA).

### 2.6. Western Blotting

The mice were perfused with 0.1 M PBS, and their tissues were rapidly dissected. The TA tissues were snap-frozen using cooled isopentane and stored at −80 °C until needed. To prepare the samples for analysis, equal amounts of total protein homogenates were loaded onto a polyacrylamide gel and electroblotted onto a PVDF membrane (Millipore, Burlington, MA, USA), following the method previously described in [25]. The membranes were then immunoblotted with primary antibodies, followed by HRP-conjugated secondary antibodies from Santa Cruz Biotechnology (Dallas, TX, USA). The Luminata Forte Western Chemiluminescent HRP Substrate from Millipore (Burlington, MA, USA) was used to develop the blots, and the Chemi-Doc XRS system from Bio-Rad (Hercules, CA, USA) was used for visualization. The immunoreactivity was normalized to the total amount of protein loaded, as determined by Ponceau staining. The primary antibodies used in the experiment were mouse anti-Pax7 (1:1000; DSHB, Iowa City, IA, USA), rabbit anti-MyoD (1:1000; Proteintech, Rosemont, IL, USA), mouse anti-MyoG (1:130; DSHB, Iowa City, IA, USA), and CD206 (1:500; Abcam, Cambridge, UK).

### 2.7. Primary Satellite Cell Cultures and Immunofluorescence

The process for isolating and labeling satellite cells was performed following the methods described in Fabbrizio et al. [25]. Hindlimb muscles were taken from three-week-old mice and digested for 45 min at 37 °C in PBS (Sigma-Aldrich, St. Louis, MO, USA) supplemented with Dispase II (2.4 U/mL, Roche, Basel, Switzerland), Collagenase A (2 mg/mL, Roche, Basel, Switzerland), 0.4 mM CaCl_2_, 5 mM MgCl_2_, and DNase I (10 μg/mL, Roche, Basel, Switzerland). The resulting cell suspension was filtered and stained with CD45/CD31/Ter119 phycoerythrin (PE) for lineage exclusion, Sca1-FITC (stem cell antigen 1), fluorescein isothiocyanate (FITC), and α7 integrin allophycocyanin (APC) APC. The SCs were then sorted using Moflo Astrios (Beckman Coulter, Brea, CA, USA), seeded onto Matrigel-coated plates (Corning, Corning, NY, USA) at a low density (3500 cells/cm^2^), and cultured for four days in Cyto-Grow complete medium (Resnova, Rome, Italy) as a growth medium (GM).

Once the cells reached confluence, they were shifted to a differentiation medium consisting of DMEM supplemented with 2% horse serum and cultured for a further 48 h to induce myogenic differentiation.

SC proliferation was evaluated for each well on image fields acquired with an Olympus virtual slide system VS110 (Olympus, Tokyo, Japan) by counting the number of DAPI+/Ki67+ (Anti-Ki67 Ab; Abcam) cells per field. SC differentiation was assessed by evaluating fiber dimension and the fusion index (%) given by the number of nuclei per myotubes stained with anti-MyHC (DSHB, Iowa City, IA, USA).

For both procedures, each well was acquired applying a stereological random sampling procedure. Briefly, a square grid of sample fields was drawn on the well image using the “Grid” function to ensure that each part of the well was equally likely to be sampled. The analysis was conducted on the average of four image fields acquired at fixed distances from each other.

### 2.8. Primary Mф Cultures

Primary Mф cultures were obtained as previously described [25]. The spleen was harvested and mechanically dissociated in RCB buffer (NH_4_Cl 150 mM, NaHCO_3_ 10 mM, and EDTA 1 mM) to obtain a single-cell suspension. The cells were then plated at a concentration of 4 × 10^6^ cells/mL in RPMI supplemented with 10% fetal bovine serum, 100 U/mL gentamycin, 100 µg/mL streptomycin, and 100 U/mL penicillin. After 2 h, non-adherent cells were removed, and the medium was enriched with 10 ng/mL mouse macrophage colony-stimulating factor (Sigma-Aldrich, St. Louis, MO, USA). After one week of culture, the cells were used for Western blotting and an immunofluorescence analysis. For MΦ polarization, the cells were stimulated for 24 h with IL-4 (PeproTech, London, UK) (10 ng/mL) or to an M2 phenotype with IL-4 (PeproTech, London, UK) (10 ng/mL) coupled with IL-10 (PeproTech, London, UK) (50 ng/mL) in the presence or absence of an Anti-IL-10 antibody (AbαIL-10; PeproTech, London, UK) (2.5 μg/mL).

### 2.9. Mф Proliferation Assay

Cell proliferation was analyzed by trypan blue exclusion. Briefly, macrophages were stimulated with IL-10 (PeproTech, London, UK) (50 ng/mL) in the presence or absence of Anti-IL-10 antibody (PeproTech, London, UK) (2.5 μg/mL). After 24 h, the cells were collected and centrifuged for 5 min at 300× *g*. The resulting pellet was resuspended in 1 mL of PBS. A solution of 0.4% trypan blue was prepared and mixed with 10 µL of the cell suspension for a manual cell count. The counting procedure was repeated at least three times.

### 2.10. Sc/Mф Co-Culture for Migration Analysis

For co-culture experiments, the SCs collected from C57-SOD1G93A mice were plated in one sulcus of the Culture-Insert 2 Well (Ibidi, Glasgow, UK), a two-well silicone insert with a defined cell-free gap that is suitable for wound healing, migration assays, 2D invasion assays, and the co-cultivation of cells. SCs were plated at a concentration of 3500 cells/cm^2^ in GM, which was replaced by DM after 24 h. The following day, the MΦs collected from the spleen of C57-SOD1G93A mice were plated in the other well at a concentration of 15,000 cells/cm^2^ in DM + M-CSF (1 μg/mL). After 24 h, the MΦs were olarized to an M2 phenotype with IL-4 (10 ng/mL) coupled with an IL-10 (50 ng/mL) treatment in the presence or absence of Anti-IL-10 antibody (2.5 μg/mL). The Culture-Insert that divided the two cultures was removed after 24 h, and the medium was changed with DM + M-CSF to maintain the differentiation of the SCs and to simultaneously support the survival of the MΦs. After 72 h, the cells were fixed with 4% paraformaldehyde and stored in PBS at 4 °C until immunohistochemical analysis.

The migration of the CD11b^+^ MΦs toward the MF20^+^ SCs was evaluated using a confocal Nikon A1 running NIS Elements (Nikon, Melville, NY, USA) at 40× magnification and by calculating the area of the sulcus separating the respective cultures with Image J (NIH, Bethesda, MD, USA). The degree of migration was extrapolated by normalizing the sulcus areas with respect to the control.

### 2.11. Statistics

Statistical analyses were performed using Prism 9.4.1 for Windows (GraphPad Software Inc., Boston, MA, USA). The mean ± SEM values were reported, with the dependent and group variables named on the y- and x-axes of the graph.

A repeated-measures ANOVA, followed by Sidak’s post-analysis, was used to analyse the parameters of disease progression in the SOD1G93A mice (body weight and PaGe test) by checking for normality in the residual and homoscedasticity, using the Geisser–Greenhouse epsilon to evaluate potential violations. The log-rank Mantel–Cox test was used to analyse symptom onset and survival length, with Kaplan–Meier plots generated. Student’s *t-*test was used for the statistical analysis of two groups, and a one-way ANOVA, followed by Fisher’s multiple comparison test, was used for more than two groups. The D’Agostino–Pearson omnibus normality test and relative QQ plots were used to assess the assumption of normality. A *p* < 0.05 was considered statistically significant for all analyses.

## 3. Results

### 3.1. Intramuscular Il-10 Administration Ameliorates the Disease Progression and Extends the Survival of C57-SOD1G93A Mice

To evaluate the effect of IL-10 on the disease progression of C57-SOD1G93A mice, we injected recombinant murine IL-10 in the hindlimb skeletal muscles, which are the primary muscle tissues affected in ALS animal models. Ten (five F and five M) and nine (five F and five M) C57-SOD1G93A mice were treated with IL-10 or PBS (vehicle) from 12 weeks of age and monitored until survival. This schedule was selected as muscle atrophy is a very early event in C57-SOD1G93A mice and is already advanced at the time of treatment.

No difference in loss of body weight was observed between the two experimental groups until the late symptomatic stage (20.5–21.5 weeks), when the IL-10 treated mice showed a higher body weight than the Vehicle mice (Figure 1A). While it excluded any significant side effect upon the induction of IL-10 in ALS mice, this result pinpointed a beneficial effect of the therapy on whole-body homeostasis.

Notably, in the IL-10 treated mice, muscle strength impairment was delayed and progressed slower than in the control group up to 22 weeks (Figure 1B). This effect was mainly due to the male mice in which the treatment was more effective than the female mice (Appendix A). The motor ability improvement of IL-10 treated mice did not translate into a significant delay in the disease onset compared to the vehicle mice (Figure 1C). Nevertheless, the gender-specific analysis revealed that transgenic males exhibited an earlier symptom onset than females (14 weeks versus 16 weeks), and IL-10 treatment delayed the inception by about two weeks compared to the control mice. (Appendix A). Finally, the IL-10 treated animals showed an increase in survival compared to the controls (vehicle = 22.7 ± 1 weeks; IL-10 = 23.7 ± 1.5 weeks) (Figure 1E). This was due to the equal contribution of female and male mice, as no significant difference in survival was registered between male or female IL-10-treated SOD1G93A mice compared to the controls (Appendix A).

### 3.2. Intramuscular Il-10 Administration Delays Muscle Atrophy in C57-SOD1G93A Mice

Based on the results, a supplementary group of nineteen SOD1G93A male mice were treated with IL-10 or PBS. The treatment began at 12 weeks of age, and the mice were sacrificed at 14 weeks of age (disease onset) (five IL-10; five Veh) and 16 weeks of age (symptomatic) (five IL-10; four Veh), following the protocol described in the previous section to evaluate ex vivo the treatment efficacy during the disease progression.

We initially evaluated the effect of in vivo IL-10 administration on the preservation of muscle mass. Consistent with the improved clinical phenotype, we observed that the IL-10-treated mice demonstrated hindlimb skeletal muscles that were less compromised than the control mice during disease progression. At 14 weeks of age, we measured muscle wasting in the QC, TA, and GCM muscles of the control mice, which were 50.9 ± 2.0%, 56.9 ± 3.4%, and 50.7 ± 2.3%, respectively, compared to their Ntg littermates. This percentage decreased to 23.04 ± 3.1%, 37.48 ± 4.3%, and 33.02 ± 3.4%, respectively, upon injection of IL-10 (Figure 2A–C). Notably, QC mass preservation was extended at the disease onset stage, with the PBS-treated C57-SOD1G93A mice showing an increase in muscle wasting of 51.1 ± 6.1% compared with the Ntg littermates. This value decreased at 38.7 ± 6.8% in the IL-10-treated C57-SOD1G93A mice (Figure 2D). Although it exhibited a similar trend to what was observed in the QC muscles, no significant change was recorded in the muscle atrophy of the TA and GCM muscles at the disease onset between the two experimental groups (Figure 2E,F).

### 3.3. Intramuscular IL-10 Administration Preserves Muscle Architecture by Eliciting Satellite Cell Proliferation and Differentiation in C57-SOD1G93A Mice

Subsequently, we compared the hindlimb muscle architecture between IL-10- and vehicle-treated C57-SOD1G93A mice at 14 weeks of age, as the treatment was more effective at this stage than at 16 weeks of age. Given the high proportion of fast-fatigable fibers (IIb) in the TA, which are affected early by the disease [7], we selected this muscle for analysis. Initially, we evaluated the effect of IL-10 administration on the differentiation rate of myofibers in the TA of both experimental groups during disease progression. We observed that the IL-10-treated mice had a higher muscle fiber cross-sectional area (CSA) than the controls (Figure 3A,B). Interestingly, an analysis of the fiber subtypes revealed a rescue of fast glycolytic IIb fibers in the IL-10-treated mice compared to the controls, with a value similar to that of the Ntg littermates (Figure 3A,C).

Next, we examined the impact of boosting IL-10 on the expression of two critical myogenic factors, Pax7, the hallmark of satellite cell (SC) stemness, and Myogenin (MyoG), a marker of early commitment and differentiation, in the TA muscles of 14 week old C57-SOD1G93A mice. We observed a significant upregulation of MyoG but not Pax7 in the TA musclse of the IL-10-treated mice compared to the control group (Figure 3D–F). Furthermore, our analysis revealed that the transcription factor critical for defining the fate of activated SCs, myoblast determination protein 1 (MyoD), was significantly upregulated upon boosting IL-10 (Figure 3D,G).

We next assessed the impact of IL-10 injections on the differentiation and proliferation rate of the C57-SOD1G93A muscle-derived myogenic precursor cells. For this purpose, a supplementary group of mice, treated with the same experimental protocol from 12 to 14 weeks, was used to set up ex vivo primary SC cultures.

Immunofluorescence against the nuclear protein Ki67 revealed a twofold higher proliferative rate of SCs challenged with IL-10 compared to the vehicle group (Figure 3H,I). Mainly, SCs sorted from the IL-10-pre-treated muscles showed a fivefold higher differentiation index than the vehicles, as assessed by the higher percentage of myocytes with two or more nuclei compared to the controls (Figure 3J,K). This evidence was corroborated in vitro by challenging the primary SC cells from untreated C57-SOD1G93A mice with IL-10 (50 ng/mL) and/or an anti-IL-10 antibody (AbαIL-10, 2.5 μg/mL) for 48 h in a differentiation medium. As a result, the IL-10-treated SCs showed a higher differentiation index than the control, and this effect was reversed upon the inhibition of IL-10 signaling through AbαIL-10 (Appendix A).

### 3.4. Intramuscular IL-10 Administration Promotes the Macrophage Polarization to M2 Pro-Regenerative Fingerprint in the Hind Limb Skeletal Muscles of C57-SOD1G93A Mice

IL-10 is a potent immunoregulator able to trigger an MΦ shift towards a pro-regenerative and anti-inflammatory M2 profile [20,26,27].

After obtaining this information, our next course of action was to explore the impact of IL-10 on MΦ density and polarization in the skeletal muscle of C57-SOD1G93A mice at 14 weeks of age. We observed that the IL-10-treated mice had a higher density of CD11b+ cells within the TA muscle compared to the control group (Figure 4A,B). This finding may suggest a direct influence of IL-10 on MΦ in situ proliferation. To verify this assumption, we assessed the in vitro proliferative index of ex vivo isolated SOD1G93A MΦs upon IL-10 administration. We found that IL-10 treated MΦs increased the proliferation index by 2.7-fold compared to the untreated MΦs, and this effect was reversed when the IL-10 effect was ablated with AbαIL-10 (Figure 4D,E).

Next, our investigation continued to focus on the inflammatory fingerprint acquired by MΦs in the skeletal muscle of C57-SOD1G93A mice when induced with IL-10. The histological analysis indicated a decrease in the density of M1 iNOS+ MΦs and a significant increase in the M2 CD206+ counterpart in the QC muscle of IL-10-treated transgenic mice compared to the controls (Figure 4A,C). An analysis of the inflammatory milieu in whole TA RNA lysates confirmed this evidence, demonstrating a significant downregulation of proinflammatory cytokines, tumor necrosis factor α (Tnfα), and Interleukin 1β (Il1-β) in association with an IL-10 increase in treated mice compared to controls as a result of a positive feedback loop (Figure 4F–H).

To validate the direct effect of IL-10 on the MΦ fingerprint, spleen-derived MΦs of C57-SOD1G93A mice were polarised toward an M2 phenotype, using IL-4 (10 ng/mL for 24 h) in the presence of IL-10 (50 ng/mL) and/or AbαIL-10 (2.5 mg/mL). The protein content of CD206 was quantified by immunoblot analysis. The co-administration of IL-4 and IL-10 showed a slight, though not significant, increase in CD206 protein levels compared to cells treated with IL-4 only (Appendix A). Additionally, IL-10 alone was unable to induce an M2 anti-inflammatory phenotype. However, the administration of AbαIL-10 strongly reduced CD206 expression in both the presence or absence of IL-10 infusion. In particular, blocking the IL-10 signaling with the AbαIL-10 antagonism limited the polarization of MΦs by IL-4, surmising that the IL-10 signaling acts synergistically with the MΦ maturation pathway to elicit the M2 phenotype (Appendix A).

### 3.5. IL-10 In Vitro Administration Enhances the Macrophage–Satellite Cell Interplay

MΦ-SC cross-talk in the damaged skeletal muscle is pivotal in supporting proper myogenesis. Following muscle injury, SCs express chemotactic factors (i.e., MCP1; VEGF) and initiate monocyte recruitment and interplay with MΦs to amplify chemotaxis and enhance muscle growth [28]. Based on this information, we investigated the potential influence of IL-10 on the MΦ migratory capability towards SCs. Primary cultures of SCs and MΦs were plated in a Culture-Insert 2 Well (Ibidi). SCs were differentiated to myotubes in DM for 48 h, while the MΦs were polarised to M2 using IL-4 (10 ng/mL) in both the presence or absence of IL-10 (50 ng/mL) and/or AbαIL-10 (2.5 mg/mL). After three days, the insert was removed to allow cell communication, evaluating the ability of the MΦs to migrate after 48 h in DM (Figure 5A).

We found that IL-10 administration alone elicited MΦ relocation even in M0-MΦ, showing a wound area close to what was observed in IL-4-treated MΦ-M2. This effect was amplified in MΦs co-treated with IL-4 and IL-10, indicating the IL-10-mediate boosting of MΦ-SC cross-talk (Figure 5A,B). Conversely, AbαIL-10 administration abolished the adjuvant effect of IL-10, restoring the migration index to the levels of untreated MΦ-M2 (Figure 5A,B).

### 3.6. Intramuscular IL-10 Administration Promotes Spinal Motor Neuron Survival and Decreases Neuroinflammation in the Spinal Cord of C57-SOD1G93A Mice

As the IL-10-treated C57-SOD1G93A mice displayed lower levels of muscle atrophy compared to the vehicle-treated mice during disease progression, we investigated whether this reduced hindlimb muscle pathology correlated with a decrease in MN loss within the CNS. We quantified large MNs with a cell body area of ≥ 400 μm^2^ after ChAT staining in the lumbar spinal cord at 14 and 16 weeks, focusing on the most vulnerable α-MNs [29]. Our results showed a significant reduction in MN loss in the lumbar spinal cord of the IL-10-treated mice compared to the controls at the onset of motor disease but not at the late presymptomatic disease stage (Figure 6A,B). Additionally, starting from the presymptomatic disease stage, the IL-10-treated mice displayed reduced astrocytosis and microgliosis compared to the vehicle-treated mice, as indicated by the lower Glial fibrillary acidic protein (GFAP) and ionized calcium-binding adapter molecule 1 (Iba1) immunoreactivity in the ventral horns of the lumbar spinal cord at both analyzed time points (Figure 6C–E). This result suggests that beneficial backward signals from skeletal muscles progressively propagate through the motor unit to protect MNs in the CNS.

### 3.7. Intramuscular IL-10 Administration Preserves Skeletal Muscle in Transgenic 129sv-SOD1G93A Mice Having Defective Immune Cell Recruitment

MΦ recruitment is an essential process for promoting myogenesis in regenerating skeletal muscle. We evaluated the effect of IL-10 administration on SC differentiation and immunomodulation in the skeletal muscle of 129Sv-SOD1G93A mice with a faster disease progression. The genetic background (129Sv) of these mice is associated with a poor ability to recruit immune cells during inflammation [12,15,30].

Ten male mice were treated with IL-10 (five) or PBS (five) at 12 weeks of age, following the protocol described above. The mice were sacrificed at 14 weeks of age.

IL-10 challenge reduced TA atrophy in transgenic mice compared to the vehicle group (Figure 7A). While they showed a degree of preservation compared to the untreated mice, no significant changes were found in muscle wasting in the QC and GCM muscles (Figure 7B,C).

As expected, the analysis of MΦ density within the TA muscles of 129Sv-SOD1G93A mice showed no significant increase in CD11b+ myeloid cells compared to the Ntg mice, and IL-10 was unable to enhance MΦ recruitment (Figure 7D,E). Nevertheless, the cytokine would influence the fingerprint of the resident myeloid population by reducing CD11b+/iNOS+ M1-MΦ density and increasing CD11b+/CD206+ M2-MΦ in the skeletal muscle of IL-10-treated mice compared to the vehicle group (Figure 7D,F). Additionally, ex vivo isolated primary SCs from IL-10-treated 129Sv-SOD1G93A mice showed enhanced differentiation in vitro (Figure 7G–H). These results suggest that in the presence of a defective immune response, IL-10 is also able, per se, to promote the generation of new myofibers through a multitarget activity towards SCs and MΦs.

## 4. Discussion

In this study, we established that the intramuscular injection of IL-10 improved the motor function of SOD1G93A mice. The beneficial effect was associated with a straight influence on the SC differentiation and MΦ polarization in the skeletal muscles, and it spread retrogradely to the spinal cord, reducing astrogliosis and preserving MNs.

Efficient SC activation, proliferation, fusion, and differentiation are crucial steps in the myogenic process that enables skeletal muscle to repair itself [16,17]. However, research on ALS mice and patients suggests that SCs may be entering a senescent state, disrupting the signaling necessary for optimal myogenesis [31]. Consequently, skeletal muscles are unable to regenerate, leading to significant muscle atrophy and weakness.

The role of the inflammatory response in muscle repair is critical in terms of both time and space as it serves as a bridge between the initial muscle injury response and muscle reparation [17,32]. Immune cells and acute inflammation are essential in almost every stage of muscle regeneration, as evidenced by data from cell biology and immunology. The inflammatory response is dynamic, with Mфs adapting their gene expression programs to control the sequential phases of tissue recovery. Following skeletal muscle injury, proinflammatory Mфs accumulate in the affected tissue area to clear debris and stimulate SC proliferation. Subsequently, these Mфs transition into anti-inflammatory Mфs, also known as pro-regenerative macrophages, which aid in the formation and growth of new myofibers [17,32].

We recently demonstrated the pivotal role of the innate, immune-mediated response in preserving skeletal muscle mass and thus the speed of the disease progression in ALS mouse models. The early, self-complementary adeno-associated virus 9 (scAAV9)-mediated monocyte chemoattractant protein 1 (MCP1) boosting in the skeletal muscle of mSOD1 mice significantly enhanced MΦ infiltration, promoting the generation of new myofibers and lessening denervation atrophy and MN loss [15]. While this approach successfully gave rise to new myofibers in transgenic mice, it did not directly influence the MΦ switch towards a pro-regenerative M2 profile or muscle stem cell differentiation. These processes occurred spontaneously during the disease progression due to the MCP1-mediated increase of MΦ recruitment. Therefore, in this study, we evaluated a preclinical therapeutic approach focused on supporting MΦ transition and triggering SC conversion to mature myofibers by exogenously administering IL-10 in the skeletal muscles of SOD1G93A mice.

IL-10 is a homodimeric type II cytokine composed of 178 amino acids. It binds a heterodimeric receptor (IL-10 receptor, IL-10-R), inducing an anti-inflammatory response [33]. The cytokine is mainly produced by antigen-presenting cells (APCs), including monocytes/MΦ, which are the main targets of IL-10 immunosuppressive effect as they express higher levels of IL-10R [20,34].

In the CSF of ALS patients with mild symptoms and a slow progressive course, IL-10 levels are elevated, suggesting a possible neuroprotective and anti-inflammatory action of this cytokine [35]. Additionally, preclinical ALS studies reported delays in symptom onset and improved survival in a SOD1G93A mouse model with lifelong AAV-IL-10 overexpression in the spinal cord [36,37]. These studies are encouraging in defining a possible therapeutic strategy for ALS. Nevertheless, the effect of IL-10 at the tissue level, especially in peripheral districts, has never been investigated.

This study demonstrated that the administration of IL-10 delayed hindlimb skeletal muscle atrophy and increased muscle fiber size by enhancing the expression of pro-differentiating factors, MyoD and MyoG. Additionally, SCs derived from the IL-10-treated muscles and cultured ex vivo displayed an improved ability to generate new mature myotubes, indicating the presence of an imprinted memory resulting from environmental cues.

### 4.1. The IL-10 Signaling Elicited the In Situ Proliferation and Polarization of MΦ toward an M2-Biased Phenotype in the Skeletal Muscle of C57-SOD1G93A Mice

IL-10’s pivotal role in myogenesis is due to its immunomodulatory effects towards APCs. In injured skeletal muscles, IL-10 is expressed by resident and infiltrating MΦs to autoregulate the inflammatory response and escort the transition of myogenesis from the proliferative stage to the differentiation/growth stage [17,27,38]. Specifically, IL-10 inhibits the infiltration of neutrophils and MΦ-M1s into the damaged tissue [39], limiting the expression of proinflammatory cytokines, and elicits the M1 to M2 shift of MΦs, a mandatory step for gaining regular muscle growth and regeneration.

In a model of chronic skeletal muscle infections, intramuscular injections of IL-10 improved muscle function and skewed MΦs toward a restorative phenotype. These shifts in the MΦ phenotype were coupled with enhanced physiologic parameters of regeneration [26]. Also, IL-10 influenced the MΦ profile during chronic muscle atrophy. Indeed, treating mdx-derived muscle MΦs with IL-10 lessened the activation of the M1-MΦs in favour of M2-MΦs [21]. Notably, IL-10 signaling was downregulated in mdx mouse skeletal muscles depleted of MΦs, while ex vivo primary SCs showed differentiation defects in myotube formation. Noteworthy, this imprinting is reversible and can be rescued in vitro by the interplay with MΦ or IL-10 challenging, surmising that MΦs recover SC differentiation defects via IL-10 signaling [24].

In keeping with this evidence, we found that IL-10 signaling influenced the peripheral immune response within the skeletal muscles of C57-SOD1G93A mice during the generation of new myofibers. It is noteworthy that the exogenous administration of IL-10 increased CD11b^+^ cell muscle density, suggesting a direct effect of the cytokine on Mфs in situ proliferation. Indeed, IL-10 administration to primary C57-SOD1G93A Mфs elicited their proliferation index in vitro. This process was previously reported by Jenkins et al. [40], who revealed that the expansion of innate cells required for pathogen control or wound repair can occur without the recruitment of tissue-destructive inflammatory cells.

Our study found that the intramuscular administration of IL-10 led to an increase in M2-MΦs within the skeletal muscles while reducing the presence of M1-Mфs and related proinflammatory factors, such as IL1β and TNFα. We further investigated the influence of IL-10 on Mфs and found that the co-administration of IL-4 and IL-10 to C57-SOD1G93A-derived Mф cultures increased the M2 profile, as evidenced by the increased expression of CD206. However, IL-10 alone was insufficient in inducing an M2 phenotype in MΦs. Conversely, in vitro polarization shift of Mфs toward the M2 phenotype was significantly suppressed following IL-10 depletion, framing the cytokine as an immunomodulator supporting but not directly eliciting the transition of MΦs to the anti-inflammatory M2 fingerprint.

Muscle IL-10 activity is not exclusive to MΦs, but the IL-10 signaling is pivotal in promoting MΦ-SCs cross-talk, an essential process in immune-cell-mediated myogenesis [17]. We found that IL-10 is crucial in mediating MΦ displacement in vitro towards SCs regardless of the degree of polarization. Although the mechanism underlying this interaction is still to be elucidated, it is likely that IL-10-treated MΦs elicited the expression of chemo-attractive factors in SCs. Indeed, following muscle insult, SCs upregulated MCP1 to initiate monocyte recruitment and interplay with MΦs to amplify chemotaxis and enhance muscle growth [28].

### 4.2. The IL-10 Signaling Triggers Satellite Cell Differentiation in an Immune-Independent Manner

In addition to influencing the inflammatory state of MΦs in the muscle, IL-10 acts directly on the fate of SCs. Indeed, IL-10 administration in primary C57-SOD1G93A SC cultures remarkably boosted their differentiation, which was fully inhibited upon IL-10 depletion. Our results corroborate previous findings that reported IL-10 ablation significantly slowed myogenesis in regenerating muscle during acute injury or chronic disease (i.e., Duchenne Muscular Dystrophy, DMD), showing a lower growth of new muscle fibers and poor resilience to damage [21,27,41]. As a counter-proof, IL-10 injections in MΦ-depleted mdx mice improved the differentiative potential of ex vivo SCs [24].

We validated this evidence in an ALS-like disease using 129Sv-SOD1G93A mice whose genetic background is associated with defective immune cell recruitment during inflammation [12,15,30]. Intramuscular IL-10 injections in these mice delayed hindlimb muscle atrophy, promoting the pro-regenerative activity of resident MΦs and enhancing the differentiation rate of ex vivo isolated SCs.

### 4.3. Preservation of Skeletal Muscle Influences Inflammation and Motor Neuron Survival within the Spinal Cord of C57-SOD1G93A Mice

Skeletal muscle plays a significant role in promoting neuron survival, axonal growth, and the maintenance of synaptic connections through anabolic signals and electric impulses. [42]. After a muscle injury, the altered activity of satellite cells (SCs) releases various cytokines/factors, which are transmitted throughout the body. [43,44,45]. These signals from newly formed myofibers and SCs might be crucial in preventing motor neuron (MN) loss. To investigate if muscle preservation impacts the protection of the entire motor unit, the study extended its analysis beyond skeletal muscles. The results showed a reduction in MN loss in the spinal cord and a decrease in astrocytosis and microgliosis throughout the progression of the disease. It is noteworthy that our approach turned out to be more effective than previous protocols in which the AAV-mediated overexpression of IL-10 in the CNS exacerbated microgliosis [36] without affecting the extent of MN loss [37].

## 5. Conclusions

In conclusion, we demonstrated that the intramuscular injection of IL-10 in SOD1G93A mice improved motor performance by decreasing muscular atrophy and mitigating CNS inflammation and MN loss.

In summary, we found that cytokine activity in the skeletal muscles of transgenic ALS mice had multiple targets. IL-10 signaling was responsible for preserving muscle fiber architecture and delaying muscle atrophy by promoting the differentiation of mature myofibers. This process was directly influenced by the effect of IL-10 on SCs and indirectly impacted by the polarization of MΦs toward an anti-inflammatory and pro-regenerative phenotype. Additionally, our findings showed that improving the condition of the skeletal muscle had a significant impact on the pathology in the CNS. This suggests that MN loss in SOD1G93A transgenic mice may be a form of retrograde, dying-back degeneration following muscle disease and is similar to human ALS [6].

Despite numerous attempts at developing therapies, the high rate of therapeutic failures has reinforced the notion that ALS is a complex disease involving various factors and systems. The disease is characterized by alterations in the structural, physiological, and metabolic parameters of MNs, glia, and muscles, which act synergistically to worsen the condition. Therefore, therapeutic strategies must target multiple mechanisms and various cells and tissues to be effective.

The encouraging results from this study have broadened our understanding of immune dynamics in the skeletal muscle of ALS mice, providing essential prospects for developing targeted immunomodulatory treatments to be used in conjunction with CNS-targeted drugs to enhance the effectiveness of potential clinical treatments. An exciting perspective could concern the long-term modulation of the peripheral immune system in the skeletal muscle with the use of AAV-mediated therapies.

Our findings afford a possible explanation for the failure of untargeted immunomodulatory treatments in ALS [46,47] and suggest that the characterization of the immune muscle profile in patients might be a clinical adjunct to improve the clinical practice and develop innovative and personalized strategies to hinder the disease progression.

## Figures and Tables

**Figure 1 cells-12-01016-f001:**
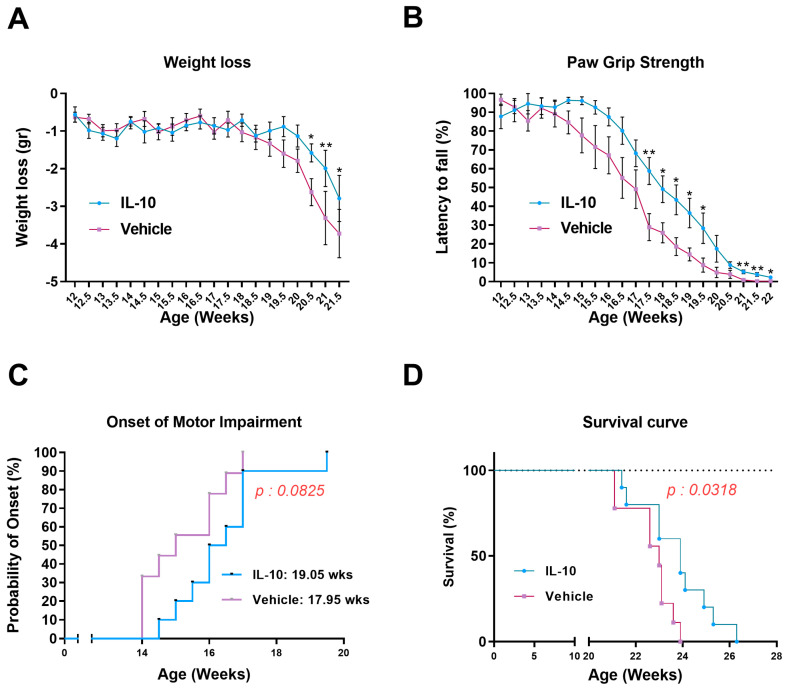
Intramuscular IL-10 administration ameliorates the disease progression and extends the survival of C57-SOD1G93A mice. (**A**) Body weight loss for IL-10-treated (*n* = 10) and PBS-treated (*n* = 9) C57-SOD1G93A mice. The data are reported as the mean ± SEM for each time point. * *p* < 0.05, ** *p* < 0.01 by repeated-measures ANOVA with Sidak’s post-analysis. (**B**) Paw grip endurance (PaGE) test for IL-10-treated (*n* = 10) and PBS-treated (*n* = 9) C57-SOD1G93A mice. The data are reported as the mean ± SEM for each time point. * *p* < 0.05, ** *p* < 0.01 by repeated-measures ANOVA with Sidak’s post-analysis. (**C**) IL-10-treated mice had a delayed onset of motor impairment compared to PBS-treated mice. *p* < 0.0825 by Mantel–Cox log-rank test. (**D**) IL-10-treated mice had a delayed end of life compared to PBS-treated mice. *p* < 0.0318 by Mantel–Cox log-rank test.

**Figure 2 cells-12-01016-f002:**
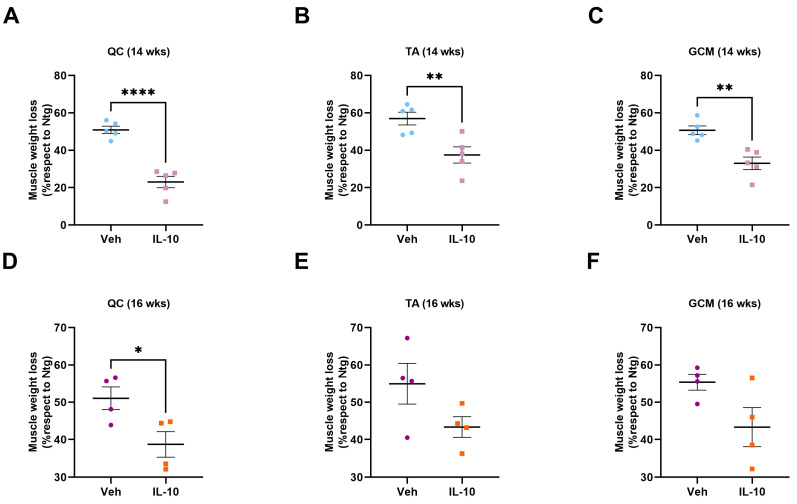
Intramuscular IL-10 administration delays muscle atrophy in C57-SOD1G93A mice. (**A–F**) Muscle wasting was calculated at 14 and 16 weeks of age by measuring the quadriceps (QC) (**A**,**D**), tibialis anterior (TA) (**B**,**E**), and gastrocnemius Medialis (GCM) (**C**,**F**) muscle weight of IL-10- and PBS-treated C57-SOD1G93A mice compared to Ntg littermates. The data are presented as th emean ± SEM. The independent experiments are scattered on the graph at each time point for each experimental group. * *p* < 0.05, ** *p* < 0.01, **** *p* < 0.0001 by unpaired *t*-test.

**Figure 3 cells-12-01016-f003:**
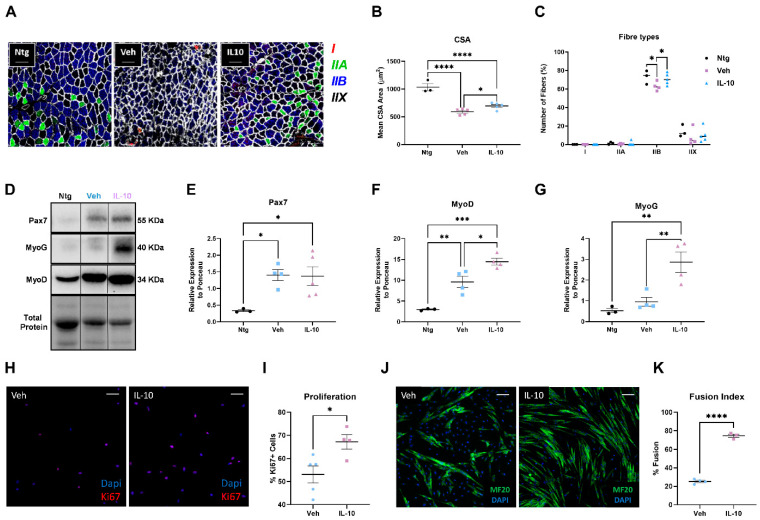
Intramuscular IL-10 administration preserves muscle architecture by eliciting satellite cell proliferation and differentiation in C57-SOD1G93A mice. (**A**) Representative confocal images showing the immunostaining of muscle fiber typing in TA muscle via myosin heavy chain (MHC) isoforms. Corresponding colour legend for fibers types: type I slow-twitch = red; type IIA fast-twitch oxidative = green; type IIX fast-twitch glycolytic = unstained (black); type IIB fast-twitch glycolytic = blue; laminin = white. Scale bar = 50 µm. (**B**,**C**) Mean cross-sectional area (CSA) (**B**) and muscle fiber composition (**C**) in IL-10-treated C57-SOD1G93A mice compared to vehicle mice at 14 weeks of age. IL-10 increases the dimension of fibers, which is associated with a shift towards muscle fibers with large size. The data are presented as (mean ± SEM) of *n* = 4–5 independent experiments for each group. * *p* < 0.05, **** *p* < 0.001 by one-way ANOVA with Tukey’s post-analysis. (**D**–**G**) Representative immunoblot images and relative densitometric analysis of Pax7 (**D**,**E**), MyoG (**D**,**F**) and MyoD (**D**,**G**) protein expression in TA muscle at 14 weeks of IL-10- and PBS-treated C57-SOD1G93A mice compared to Ntg littermates. The data are presented as (mean ± SEM) of *n* = 4–5 independent experiments for each group. * *p* < 0.05, ** *p* < 0.01, *** *p* < 0.001 by one-way ANOVA with Tukey’s post-analysis. (**H**) Representative confocal images showing the immunostaining for Ki67 (red) and DAPI (blue) on C57-SOD1G93A SCs derived from pre-treated muscles with IL-10 or vehicle for two weeks and cultured in GM for four days. Scale bar = 100 µm. (**I**) The proliferation index was assessed by simultaneously counting the number of Ki67 and DAPI juxtapositions in primary C57-SOD1G93A SC cultures. * *p* < 0.05 by unpaired *t*-test. (**J**) Representative confocal images showing the immunostaining for MF20-MyHC (green) and DAPI (blue) on C57-SOD1G93A SCs derived from pre-treated muscles with IL-10 or vehicle for two weeks and cultured in DM for 48 h. Scale bar = 100 µm. (**K**) The fusion index was calculated as (number of nuclei in MyHC+ cells with ≥2 nuclei/total number of nuclei). Data are reported by means ± SEM from at least three independent experiments for each group. **** *p* < 0.0001 by unpaired *t*-test.

**Figure 4 cells-12-01016-f004:**
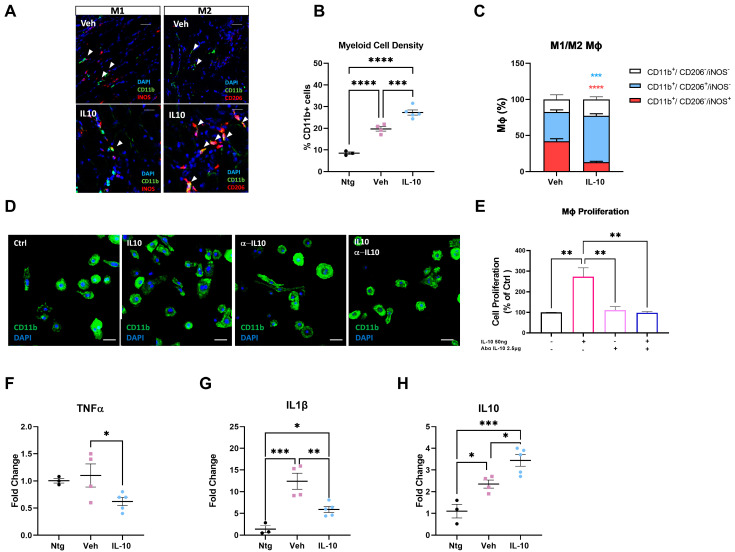
Intramuscular IL-10 administration promotes macrophage polarization to M2 pro-regenerative fingerprint in the skeletal muscles of C57-SOD1G93A mice. (**A**) Representative confocal images showing the distribution of Mф-M1s (CD11b+/iNOS+) and Mф-M2s (CD11b+/CD206+) in longitudinal TA muscle sections of C57-SOD1G93A mice treated once daily with IL-10 or PBS at 12 weeks of age for 15 days (scale bar = 100 µm). (**B**,**C**) Quantification of MΦ infiltration (**B**) and fingerprint of Mф-M1s (CD11b+/iNOS+) and Mф-M2 (CD11b+/CD206+) (**C**) within the TA of IL-10 and vehicle-treated groups. The data are reported as (mean ± SEM) from at least five independent experiments for each group. *** *p* < 0.001, **** *p* < 0.0001 by one-way ANOVA with Tukey’s post-analysis. (**D**) Representative confocal images showing the proliferation of Mфs stimulated with IL-10 in the presence or absence of anti-IL-10 antibody (scale bar = 50 µm). (**E**) Quantification of Mф proliferation by trypan blue exclusion. The data are reported as (mean ± SEM) from at least three independent experiments. ** *p* < 0.01 by one-way ANOVA with Tukey’s post-analysis. (**F**–**H**) Real-time qPCR for TNFα (**F**), IL-1β (**G**), and IL-10 (**H**) mRNA transcripts in TA muscles at 14 weeks of IL-10 and PBS-treated C57-SOD1G93A mice compared to Ntg littermates. The data are presented as (mean ± SEM) of *n* = 4–5 independent experiments for each group. The data are reported as (mean ± SEM). * *p* < 0.05, *** *p* < 0.001 by one-way ANOVA with Tukey’s post-analysis.

**Figure 5 cells-12-01016-f005:**
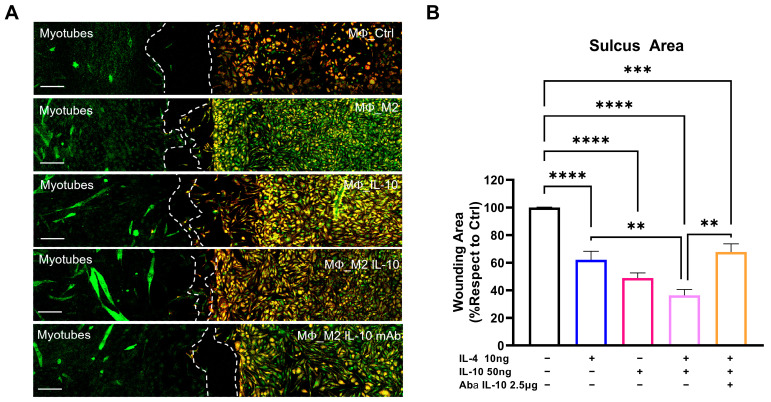
IL-10 in vitro administration enhances the macrophage–satellite cell interplay. (**A**) Representative confocal images showing the immunostaining for primary C57-SOD1G93A-derived SCs (MF20-MyHC, green) and MΦs (CD11b, yellow). MΦs were selectively polarised to the M2 phenotype and treated with 50 ng/mL IL-10. Dotted lines spotlight the area of sulcus at 48 h following the removal of the insert. Scale bar = 100 μm. (**B**) The migration rate was calculated using the wound-healing assay. The data are reported as (mean ± SEM) from at least six independent experiments. ** *p* < 0.01, *** *p* < 0.001, **** *p* < 0.0001 by one-way ANOVA with Tukey’s post-analysis.

**Figure 6 cells-12-01016-f006:**
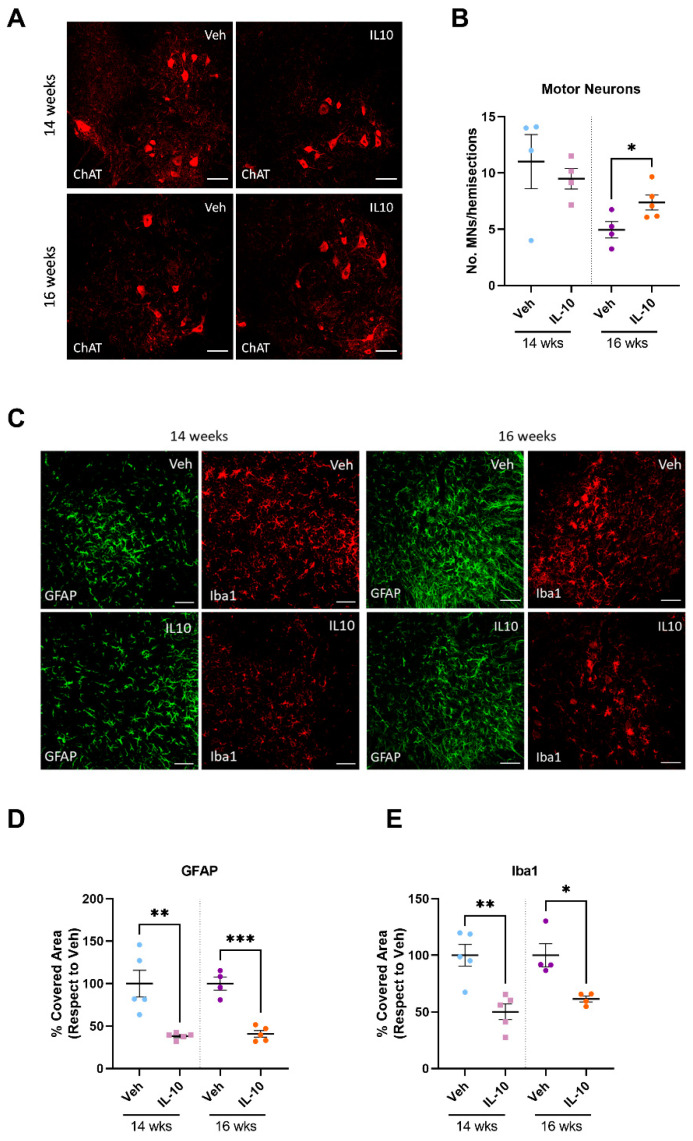
Intramuscular IL-10 administration promotes spinal motor neuron survival and decreases neuroinflammation in the spinal cord of C57-SOD1G93A mice. (**A**) Representative confocal images of Chat-immunostained lumbar spinal cord transverse sections of IL-10- and PBS-treated C57-SOD1G93A mice at 14 and 16 weeks of age. Scale bar = 100 µm. (**B**) Quantification of MNs Chat+ in the spinal cords of IL-10- and PBS-treated C57-SOD1G93A mice at 14 weeks of age. The data are expressed as mean ± SEM of MNs (≥ 400 μm^2^) per hemisection. Each symbol in the graph is the average quantification of at least ten serial sections (i.e., 10 hemisections) for each animal. * *p* < 0.05 by unpaired *t*-test. (**C**) Representative confocal images showing GFAP (green) and Iba1 (red) staining in lumbar spinal cord transverse sections of IL-10- and PBS-treated C57-SOD1G93A mice at 14 and 16 weeks of age. Scale bar = 100 µm. (**D**,**E**) Quantification of the percentage of covered area for GFAP (**D**) and Iba1 (**E**) immunostaining in the spinal cords of IL-10- and PBS-treated C57-SOD1G93A mice at 14 weeks of age. The data are expressed as mean ± SEM. * *p* < 0.05, ** *p* < 0.01, *** *p* < 0.001 by unpaired *t*-test. Each symbol in the graph is the average quantification of at least five sections (i.e., No. 10 hemisections) for each animal.

**Figure 7 cells-12-01016-f007:**
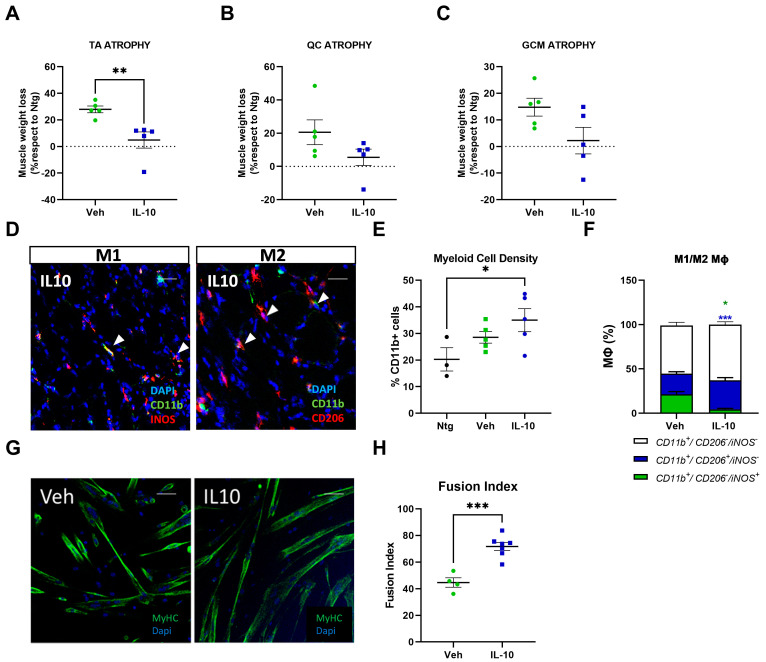
Intramuscular IL-10 administration preserves skeletal muscle in transgenic 129Sv-SOD1G93A mice with defective immune cell recruitment. (**A**–**C**) Muscle wasting was calculated at 14 weeks of age by measuring the tibialis anterior (TA) (**A**) auadriceps (QC) (**B**) and gastrocnemius medialis (GCM) (**C**) muscle weight of IL-10- and PBS-treated 129Sv-SOD1G93A mice compared to their Ntg littermates. The data are presented as mean ± SEM. The independent experiments are scattered on the graph at each time point for each experimental group. ** *p* < 0.01 by unpaired *t*-test. (**D**) Representative confocal images showing the distribution of MΦ-M1s (CD11b+/iNOS+) and MΦ-M2s (CD11b+/CD206+) in longitudinal TA muscle sections of 129Sv—SOD1G93A mice treated once daily with IL-10 or PBS at 12 weeks of age for 15 days (scale bar = 100 µm). (**E**,**F**) Quantification of MΦ infiltration (**E**) and fingerprint of MΦ-M1s (CD11b+/iNOS+) and MΦ-M2s (CD11b+/CD206+) (**F**) within the TA of IL-10 and vehicle-treated groups. The data are reported as (mean±SEM) from at least five independent experiments for each group. * *p* < 0.05, *** *p* < 0.001, by unpaired *t*-test. (**G**) Representative confocal images showing the immunostaining for MF20-MyHC (green) and DAPI (blue) on 129Sv-SOD1G93A SCs derived from muscles pre-treated with IL-10 or vehicle for two weeks and cultured in DM for 48 h. Scale bar = 50 µm. (**H**) The fusion index was calculated as (No. nuclei present in MyHC+cells with two or more nuclei/No. myotubes). Data are reported as the mean ± SEM from three independent experiments for each group. *** *p* < 0.001 by unpaired *t*-test.

## Data Availability

Data, materials, and software information supporting the conclusions of this article are included within the present article and its Appendix A.

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
