# Peer review of "Intramuscular IL-10 Administration Enhances the Activity of Myogenic Precursor Cells and Improves Motor Function in ALS Mouse Model"

_cells, 2023, doi:10.3390/cells12071016_

Round 1

Reviewer 1 Report

The manuscript reports the beneficial effects of intramuscular injections of IL-10 on amyotrophic lateral sclerosis (ALS) disease, using two transgenic mice models, C57-SOD1G93A and 129Sv-SOD1G93A. The benefits were observed in the onset of the disease and its progression, and in survival rate. Muscle atrophy is less severe in IL-10-injected animals which is associated with larger size of muscle fibers. In vitro myogenesis is also promoted by IL-10. At the level of macrophages, IL-10 promotes their muscle infiltration and favors their switch towards the M2 pro-regenerative profile. Interestingly, muscle IL-10 injection confers a protective effect on motor neuron decline and decreases astrogliosis in the spinal. The manuscript is straightforward, well presented and provides valuable insights into the positive effects of IL-10 on ALS disease. Some points would however require clarification.

Figure 2: Can the authors provide a tentative explanation as to why the beneficial effect of IL-10 does not last?

Figure 3: It would help the understanding to explicitly state that the fibers IIx actually are the one negative for all three antibodies tested (I, IIA and IIB). Panel A, the scale bar is not visible.

Panel K: the explanation of the fusion index in the legend is not clear. To my understanding, it should be modified as: the number of nuclei in MyHC+ cells with > 2 nuclei / total number of nuclei. Please verify.

Figure 4: It should be clearly stated which markers (CD11b, iNOS and CD206) allow the separation between M1 and M2 macrophages. It is not straightforward for non-specialists.  

Figure 4D, the labelling for the staining is poorly visible.

Figure 5: What is the green staining on the right side of the image, corresponding to macrophages? CD11b is in yellow, so what is green?

To write SC (satellite cells) for the side where culture is differentiated is misleading. These are actually myotubes and reserve cells, not any more SC. In the discussion related to this experiment (lines 613-621), the authors write that “macrophages treated with IL-10 upregulated the expression of chemo-attractive factors and elicited their induction in SCs”. This is not clear, in the experiment the macrophages are moving towards the differentiated myotubes and reserve cells, not the opposite. The interpretation of these results is not convincing at this stage.

Figure 6: To help the readers that are not in the field, it would be useful to explain the GFAP and Iba1 staining. What cells are identified by this staining? To my understanding Iba1 stains microglial cells, so is the term astrogliosis appropriate? It would help to define the term more precisely.

Figure 6 legend: It is always written that the results were obtained at 14 weeks, while all panels reported 14 and 16 weeks. Please correct.

Phlogosis: could you provide a definition for non-specialists?

Author Response

Dear Dr Man,

thank you for the professional handling and the Reviewer's constructive comments on our manuscript entitled "Intramuscular IL-10 administration enhances myogenesis and improves motor function in ALS mouse models".

Following these comments and the "ithenticate" report, we have revised our manuscript consistently, highlighting in tracking mode all the amendments.

The main corrections and responses to the Reviewer are listed below.

Reviewer 1

The manuscript reports the beneficial effects of intramuscular injections of IL-10 on amyotrophic lateral sclerosis (ALS) disease, using two transgenic mice models, C57-SOD1G93A and 129Sv-SOD1G93A. The benefits were observed in the onset of the disease and its progression, and in survival rate. Muscle atrophy is less severe in IL-10-injected animals which is associated with larger size of muscle fibers. In vitro myogenesis is also promoted by IL-10. At the level of macrophages, IL-10 promotes their muscle infiltration and favors their switch towards the M2 pro-regenerative profile. Interestingly, muscle IL-10 injection confers a protective effect on motor neuron decline and decreases astrogliosis in the spinal. The manuscript is straightforward, well presented and provides valuable insights into the positive effects of IL-10 on ALS disease. Some points would however require clarification.

We thank the Reviewer for the positive feedback on our work.

Figure 2: Can the authors provide a tentative explanation as to why the beneficial effect of IL-10 does not last?

ALS is a multifactorial disorder in which several pathological processes contribute to its pathogenesis and progression. The IL-10 treatment delayed the disease course by boosting the myogenic activity of SCs and counteracting muscle wasting. However, as the disease progresses, other pathological factors at the central and peripheral levels contribute to exhausting the effect of the treatment, making it ineffective.

Figure 3: It would help the understanding to explicitly state that the fibers IIx actually are the one negative for all three antibodies tested (I, IIA and IIB). Panel A, the scale bar is not visible.

We add the following sentence in the caption of Figure 3:

"Representative confocal images showing the immunostaining of muscle fibre typing via myosin heavy chain (MHC) isoforms. Corresponding colour legend for fibres types: type I slow-twitch = red, type IIA fast-twitch oxidative = green, type IIX fast-twitch glycolytic = unstained (black), type IIB fast-twitch glycolytic = blue, and laminin = white."

For each image of panel A, the scale bar is present under the headings (Ntg; Veh; IL10).

Panel K: the explanation of the fusion index in the legend is not clear. To my understanding, it should be modified as: the number of nuclei in MyHC+ cells with > 2 nuclei / total number of nuclei. Please verify.

We apologize to the Reviewer for the mistake. We have changed the legend of the figure accordingly.

Figure 4: It should be clearly stated which markers (CD11b, iNOS and CD206) allow the separation between M1 and M2 macrophages. It is not straightforward for non-specialists. 

We agree with the Reviewer. Two headers have been placed above the images of panels 4A and 7D to make the separation between M1 and M2 macrophages more understandable. 

Figure 4D, the labelling for the staining is poorly visible.

We thank the Reviewer for the notification; we have enlarged and contrasted the labellings of Figure 4D.

Figure 5: What is the green staining on the right side of the image, corresponding to macrophages? CD11b is in yellow, so what is green?

We thank the Reviewer for notifying this issue. The green in the macrophage section is an artefact due to the 488-FITC signal entering and overlapping with the 647-Cy5 channel (SEE THE IMAGE IN THE PDF ENCLOSED). The latter is indeed red, but it overlaps with the green signal becoming yellow during the acquisition. For this reason, the macrophage staining was definide as yellow in the caption. Despite having acquired with the confocal microscope, we could not separate the channels sufficiently to isolate the two immunostainings. However, we are sure of the cell typology for each section, as the SCs and macrophages were plated in the presence of the insert and are morphologically different.

To write SC (satellite cells) for the side where culture is differentiated is misleading. These are actually myotubes and reserve cells, not any more SC. In the discussion related to this experiment (lines 613-621), the authors write that "macrophages treated with IL-10 upregulated the expression of chemo-attractive factors and elicited their induction in SCs". This is not clear, in the experiment the macrophages are moving towards the differentiated myotubes and reserve cells, not the opposite. The interpretation of these results is not convincing at this stage.

We thank the Reviewer for raising this point; the term SC was replaced with myotubes, and the highlighted paragraph in the Discussion has been modified as follows:

"Although the mechanism underlying this interaction is still to be elucidated, it is likely that IL-10-treated MΦ elicited the expression of chemo-attractive factors in SCs. Indeed, following muscle insult, SCs upregulated MCP1 to initiate monocyte recruitment and interplay with MΦ to amplify chemotaxis and enhance muscle growth".

Figure 6: To help the readers that are not in the field, it would be useful to explain the GFAP and Iba1 staining. What cells are identified by this staining? To my understanding Iba1 stains microglial cells, so is the term astrogliosis appropriate? It would help to define the term more precisely.

We agree with the Reviewer that the term astrogliosis can be misleading. Therefore, in the Results, we spotlighted the difference between microgliosis and astrocytosis and the related markers as follows:

"Additionally, starting from the presymptomatic disease stage, IL-10-treated mice displayed reduced astrocytosis and microgliosis compared to vehicle-treated mice, as indicated by the lower Glial fibrillary acidic protein (GFAP) and Ionized calcium-binding adapter molecule 1 (Iba1) immunoreactivity in the ventral horns of the lumbar spinal cord at both analysed time-points."

Figure 6 legend: It is always written that the results were obtained at 14 weeks, while all panels reported 14 and 16 weeks. Please correct.

We thank the Reviewer for the notification. The opportune changes have been addressed.

Phlogosis: could you provide a definition for non-specialists?

Phlogosis is synonymous for inflammation. To be more straightforward, we replace the term with "inflammation" in the new version of the manuscript.

Reviewer 2 Report

Fabbrizio and colleagues have contributed a manuscript that describes work showing that intramuscular IL-10 is therapeutic in a genetic (mutant G93A-SOD1) model of ALS in mice. They characterize the effects of IL-10 on skeletal muscle satellite cells and macrophages.  

The work has many strengths. This multisystem disease and therapeutic approach integrating the CNS, skeletal muscle, and the immune system showcases beautifully the biology and complexity of the disease. Moreover, a variety of approaches is used including animal and cell culture. The study is designed, approached, and written logically and clearly.

The work has some weaknesses that could be addressed.

11.       The authors should provide a contemporary justification for using the standard G93A-SOD1 mouse model. Yes, it is the most widely used mouse, but nothing has translated to the clinics with the 25 year’s worth of use with this model. Please explain what could be different here.

22.       The injection of different muscles is a fantastic design. With the daily injections, did injection site problems and infection present?

33.       Please supply information on the purity of the IL-10.

44.       Please describe the negative controls for the immunostaining of cell cultures and tissue sections.

55.       The statistical analyses are generally good. However, the study is underpowered to make conclusions on male and female differences (first sentence in Discussion)

66.        Figure 3. A) For Fig. 3A please explain the different fiber-type markers. B) In Fig. 3H the plating densities appear to be very different in veh versus IL-10 cultures based on the DAPI staining. C) There is no description in the methods on the counting approach. Perhaps values are all normalized to total DAPI cells but there is no marker for satellite cells in this experiment. D) In Fig. 3J the culture cell plating’s and growths look very different. Veh culture is very disorganized, and the IL-10 culture is organized in pattern. Is this biological of technical?  

77.       Figure 4. For Fig. 4A, it seems like images from PBS-treated mice are missing. Also, the images and analysis need to consider relationships to blood vessels in the field of counting.

88.       The experiments and the data for Figure 5 need to be much better explained. As it is, Fig. 5A is not convincing in should major differences with any treatments. Why are experiments not done with cells collected from WT spleens and skeletal muscles? How was the quantification done?

99.       Figure 6. You cannot determine causality from this experiment, e.g., decreasing spinal cord inflammation protects motor neurons. Also, there need for care taken to show in Fig. 6A that the images shown are taken from similar spinal cord levels.  

110.   References 7 and 8 are duplicates.    

Author Response

Dear Dr Man,

thank you for the professional handling and the Reviewer's constructive comments on our manuscript entitled "Intramuscular IL-10 administration enhances myogenesis and improves motor function in ALS mouse models".

Following these comments and the "ithenticate" report, we have revised our manuscript consistently, highlighting in tracking mode all the amendments.

The main corrections and responses to the Reviewer are listed below.

Reviewer 2

Fabbrizio and colleagues have contributed a manuscript that describes work showing that intramuscular IL-10 is therapeutic in a genetic (mutant G93A-SOD1) model of ALS in mice. They characterise the effects of IL-10 on skeletal muscle satellite cells and macrophages.  

The work has many strengths. This multisystem disease and therapeutic approach integrating the CNS, skeletal muscle, and the immune system showcases beautifully the biology and complexity of the disease. Moreover, a variety of approaches is used including animal and cell culture. The study is designed, approached, and written logically and clearly.

We thank the Reviewer for the positive feedback on our work.

The work has some weaknesses that could be addressed.

  1. The authors should provide a contemporary justification for using the standard G93A-SOD1 mouse model. Yes, it is the most widely used mouse, but nothing has translated to the clinics with the 25 year's worth of use with this model. Please explain what could be different here.

We thank the Reviewer for raising this point. This study is proof of concept to establish the relevance of a muscle-related biological process never afforded in ALS. The mSOD1 mouse was essential because skeletal muscle dysfunction is well-characterised in this model compared to others.

We agree with the Reviewer that using mSOD1 mice has not yet led to identifying an effective therapy in humans. However, the same holds for animal models with other ALS-related mutations (e.g., TDP43; Fus; C9ORF), which, in addition to being less characterised, do not can still mimic the disease as accurately as the mSOD1 mouse.

Over the years, mSOD1 murine models have enabled the identification of various pathological processes underlying the disease pathogenesis. In addition to identifying widely recognised pathological mechanisms such as protein aggregation, mitochondrial dysfunction and oxidative stress, the mSOD1 model has allowed us to classify ALS as a non-cell autonomous disease in which the contribution of non-neuronal cells plays a decisive role in the pathology development. Starting from these bases, further studies have been developed in mSOD1, which have led to classifying relevant targets such as Treg cells, microglia and myofibers. This has allowed the development of clinical trials currently underway (COURAGE-ALS; Trimetazidine; IL-2; Masitinib).

That said, given the promising results obtained, we are considering extending the study to other ALS murine models (PrP-hFUS; Prp-TDP43Q331K mice) to strengthen our original hypothesis and increase its translational potential.

  1. The injection of different muscles is a fantastic design. With the daily injections, did injection site problems and infection present?

We appreciate the Reviewer for acknowledging and recognising the importance of the specific route of administration. Throughout the treatment process, we did not experience any issues or difficulties. We followed animal welfare guidelines, and the administration of 10 µL into the muscle was well-tolerated by the mice. This route of administration is routinely used in our lab (Ref. 27 in the Ms) and is safe for animals.

  1. Please supply information on the purity of the IL-10.

The Prepotech data sheet describes the compound as follows: "Purity: ≥ 98% by SDS-PAGE gel and HPLC analyses". We attach product links https://www.peprotech.com/en/recombinant-murine-il-10  

  1. Please describe the negative controls for the immunostaining of cell cultures and tissue sections.

A brief description has been added to the material and methods, section 2.4, of the new version of the manuscript as follows:

"All the immunofluorescences were verified with control experiments using only the secondary antibody."

  1. The statistical analyses are generally good. However, the study is underpowered to make conclusions on male and female differences (first sentence in Discussion)

We agree with the Reviewer that analysing a small group of mice could lead to misinterpretations. For this reason, analyses of gender-specific behaviour were included as supplementary material. Based on the criticism raised, we have removed claims of a specific gender effect from the discussion and conclusions.

  1. Figure 3. A) For Fig. 3A please explain the different fiber-type markers. B) In Fig. 3H the plating densities appear to be very different in Veh versus IL-10 cultures based on the DAPI staining. C) There is no description in the methods on the counting approach. Perhaps values are all normalised to total DAPI cells but there is no marker for satellite cells in this experiment. D) In Fig. 3J the culture cell plating's and growths look very different. Veh culture is very disorganised, and the IL-10 culture is organised in pattern. Is this biological of technical?  

We are sorry for omitting relevant details for the chomprension of data.

A. We add the following sentence in the caption of Figure 3:

"Representative confocal images showing the immunostaining of muscle fibre typing via myosin heavy chain (MHC) isoforms. Corresponding colour legend for fibres types: type I slow-twitch = red, type IIA fast-twitch oxidative = green, type IIX fast-twitch glycolytic = unstained (black), type IIB fast-twitch glycolytic = blue, and laminin = white."

B. Figure 3H is a representative image of the increased proliferation index of isolated SC cells after the IL-10 treatment. Rather than a higher plating density, it shows a higher number of DAPI/Ki67 colocalisation. SCs were sorted, ensuring accurate cell counts that allowed plating at the same density (3500 cells/cm2) Veh and treated cells in 24 multi wells plates.

C. In the materials and methods, section 2.7, we added the sentence describing the quantification of the extent of proliferation and differentiation:

"SC proliferation was evaluated for each well on stereological image fields acquired with an Olympus virtual slide system VS110 (Olympus) by counting the number of DAPI+/Ki67+ (Anti-Ki67 Ab; Abcam) cells per field. SC differentiation was assessed by evaluating fibre dimension and the fusion index (%) given by the number of nuclei per myotubes stained with anti-MyHC (DSHB)."

D. The disorganisation of the fibres in Figure 3J is associated with a lower differentiation index in Veh than IL-10 SCs. Indeed, while Vehicle mice have many satellite cells in an undifferentiated state (DAPI staining only), IL10-treated mice show a higher number of cells positive for the MF20 marker, indicating a higher number of myotubes formed.

The statistic refers to biological replicates. Being cells extracted from adult-treated animals, each animal corresponds to a biological sample.

  1. Figure 4. For Fig. 4A, it seems like images from PBS-treated mice are missing. Also, the images and analysis need to consider relationships to blood vessels in the field of counting.

Based on the request, representative images of the vehicle have been added to Figure 4A. 

We also thank the Reviewer for the alert on blood vessels. As described in the paragraph "Materials and Methods" Page 3, lane 133, all the mice used for the analyses of this work were perfused in PBS to overcome this issue.

  1. The experiments and the data for Figure 5 need to be much better explained. As it is, Fig. 5A is not convincing in should major differences with any treatments. Why are experiments not done with cells collected from WT spleens and skeletal muscles? How was the quantification done?

The following paragraph was added in section 2.10 of the Materials and Methods:

"Migration of CD11b+ MФ towards MF20+ SCs was evaluated with confocal Nikon A1 running NIS Elements (Nikon) at 40X magnification by calculating the area of the sulcus separating the respective cultures with Image J (NIH). The degree of migration was extrapolated by normalising the sulcus areas with respect to control."

In addition, the following sentence was added to the caption of Figure 5:

 "Dotted lines spotlight the area of sulcus at 48h following the remotion of the insert."

Besides, the term "wound healing" was removed from the Results description and Figure 5B as it is misleading in the context.

Experiments were exclusively done on SOD1G93A mice-derived cells to mimic in vivo conditions, where only transgenic mice were treated.

  1. Figure 6. You cannot determine causality from this experiment, e.g., decreasing spinal cord inflammation protects motor neurons. Also, there need for care taken to show in Fig. 6A that the images shown are taken from similar spinal cord levels.

We agree with the Reviewer that there may be no causality between the two events. For this reason, we changed the header of the Results section, the Figure caption and the discussion of the result as follows:

Results/Figure 6: "Intramuscular IL-10 administration promotes spinal motor neuron survival and decreases neuroinflammation in the spinal cord of C57-SOD1G93A mice."

Discussion: "We found a reduced MN loss in the spinal cord and decreased astrocytosis and microgliosis across the disease progression"

  1. References 7 and 8 are duplicates.    

         We are sorry for the mistake. We have amended the References section.

Reviewer 3 Report

I think the authors have done important work and the results are interesting. However, throughout the study, several concepts (regeneration, repair, myogenesis, etc.) are used inappropriately. This generates, in my view, a great deal of confusion when reading the manuscript. In my opinion, this should be clarified.

In the present study, the authors find that the injection of I-10 counteracts muscle atrophy. For example:

-    (Page 17, lines 652-654) that intramuscular injection of IL-10 in SOD1G93A mice improved motor performance by decreasing muscle atrophy and mitigating astrogliosis and motor neuron loss.

- (Page 16, lines 570-572) “In the present study, we showed that IL-10 administration delayed the atrophy of hindlimb skeletal muscles, which was associated with increased muscle fiber size due to an enhanced expression of pro-differentiating factors, MyoD and MyoG.”

Inflammation is recognized to play a role in ALS-related motor neuron pathology and has been shown to accompany motor neuron degeneration in the central and peripheral nervous systems of rodent models of ALS and human patients (Van Dyke et al 2016, doi:10.1016/j.expneurol.2016.01.008); they have shown that activated inflammation and abnormal glial responses occur in the limb muscle of familial ALS model rats (SOD1G93A), specifically near denervated neuromuscular junctions (NMJs).

It is well known that the alterations that occur in skeletal muscle in these situations give rise to muscle atrophy, but not to muscle fiber necrosis and, therefore, to the possible regenerative response. The regeneration of muscle fibers is a consequence of their necrosis. For this reason, the authors talk about regeneration in the wrong way. Recovery from atrophy is not regeneration.

It is well known that the loss or alteration of the innervation in the muscle fibers causes the response of the population of satellite cells and, therefore, the overexpression of the regulatory factors of myogenesis. But the activation, proliferation, and differentiation of satellite cells do not necessarily imply "regeneration" (if there is no necrosis of muscle fibers). Are the authors referring to the regeneration of the NMJS?

Another questions:

-        I believe that the procedure to evaluate the effects of IL10 on muscle atrophy should have been done by systemic injection and not intramuscular. What is the reason for injecting IL-10 intramuscularly and not systemically?

-        If the authors inject IL10 intramuscularly, they should know that the needle itself causes the destruction of muscle fibers and, consequently, a subsequent regenerative response. Is this implicated in the myogenic response observed by the authors to the administration of IL-10?

-        The authors point out that they treat and extract three muscles (Tibialis Anterior (TA), Gastrocnemius Medialis (GCM), and Quadriceps (QC)) (Page 3, line 130). However, they only refer to the TA muscle in the Immunohistochemistry procedures (page 3, lines (138-141), Western blotting (page 4, lines 185-187),...

-        The authorizations for the study by the agencies are repeated (pages 2-3, lines 99-105) (page 3, lines 120-128).

-        The abstract should be rewritten.

Author Response

Dear Dr Man,

thank you for the professional handling and the Reviewer's constructive comments on our manuscript entitled "Intramuscular IL-10 administration enhances myogenesis and improves motor function in ALS mouse models".

Following these comments and the "ithenticate" report, we have revised our manuscript consistently, highlighting in tracking mode all the amendments.

The main corrections and responses to the Reviewers are listed below.

Reviewer 3

I think the authors have done important work and the results are interesting. However, throughout the study, several concepts (regeneration, repair, myogenesis, etc.) are used inappropriately. This generates, in my view, a great deal of confusion when reading the manuscript. In my opinion, this should be clarified. In the present study, the authors find that the injection of IL-10 counteracts muscle atrophy. For example:

- (Page 17, lines 652-654) that intramuscular injection of IL-10 in SOD1G93A mice improved motor performance by decreasing muscle atrophy and mitigating astrogliosis and motor neuron loss.

- (Page 16, lines 570-572) "In the present study, we showed that IL-10 administration delayed the atrophy of hindlimb skeletal muscles, which was associated with increased muscle fiber size due to an enhanced expression of pro-differentiating factors, MyoD and MyoG."

Inflammation is recognised to play a role in ALS-related motor neuron pathology and has been shown to accompany motor neuron degeneration in the central and peripheral nervous systems of rodent models of ALS and human patients (Van Dyke et al 2016, doi:10.1016/j.expneurol.2016.01.008); they have shown that activated inflammation and abnormal glial responses occur in the limb muscle of familial ALS model rats (SOD1G93A), specifically near denervated neuromuscular junctions (NMJs).

It is well known that the alterations that occur in skeletal muscle in these situations give rise to muscle atrophy, but not to muscle fiber necrosis and, therefore, to the possible regenerative response. The regeneration of muscle fibers is a consequence of their necrosis. For this reason, the authors talk about regeneration in the wrong way. Recovery from atrophy is not regeneration.

It is well known that the loss or alteration of the innervation in the muscle fibers causes the response of the population of satellite cells and, therefore, the overexpression of the regulatory factors of myogenesis. But the activation, proliferation, and differentiation of satellite cells do not necessarily imply "regeneration" (if there is no necrosis of muscle fibers). Are the authors referring to the regeneration of the NMJs?

The terms "myogenesis" and "muscle regeneration" used in the paper refer to the myogenic precursor cell (satellite cells; SCs) response (activation, proliferation, and differentiation) and are concepts frequently used in chronic muscle atrophy (e.g., Duchenne, sarcopenia, Cachexia) (Madaro et al., 2019, PMID: 31626629; Hauerslev et al., 2014, PMID: 24963862; Arneson et al., 2019, PMID: 31706505; Huo et al., 2022, PMID: 36035464) or denervation atrophy (e.g., ALS) (Tsitkanou et al., 2016, PMID: 27679581). In this context, muscle regeneration or myogenesis is intended as the SC activity essential to produce new myofibers that replace the atrophied ones (myogenesis of new myofibres). According to recent reports, this process is also essential to promote the expression and clusterisation of AChR subunits in brand-new NMJs.

We recognize that this terminology may appear misleading in the manuscript. Therefore, to meet the Reviewer's request, where opportune, we have replaced the terms repair, myogenesis, and muscle regeneration with "generation of new myofibres" or differentiation of new myofibres.

Regarding the work of Van Dyke et al., cited by the Reviewer, it is purely descriptive, highlighting the presence of macrophages and an increase in inflammatory factors in the skeletal muscle of SOD1G93A rats assuming that this process is detrimental to the disease. Instead, we recently demonstrated how the early self-complementary adeno-associated virus 9 (scAAV9)-mediated Monocyte Chemoattractant Protein 1 (MCP1) boosting in the skeletal muscle of mSOD1 mice significantly improved motor performance by enhancing MΦ infiltration, promoting the formation of new myofibres, lessening denervation atrophy and MN loss (Ref. 17 in the Ms).

Another questions:

- I believe that the procedure to evaluate the effects of IL10 on muscle atrophy should have been done by systemic injection and not intramuscular. What is the reason for injecting IL-10 intramuscularly and not systemically?

This study is proof of concept to establish the relevance of a muscle-related biological process never afforded in ALS. IL-10 was injected intramuscularly to ensure a valuable therapeutic index (high local cytokine concentration; low systemic exposure). However, we are aware of the limits of this approach in a clinical perspective. In this respect, other routes of administration or other methodological approaches will be necessary. Indeed, in the conclusions, we highlighted the following sentence as a perspective:

"An exciting perspective could concern the long-term modulation of the peripheral immune system in the skeletal muscle with the use of AAV-mediated therapies".

- If the authors inject IL10 intramuscularly, they should know that the needle itself causes the destruction of muscle fibers and, consequently, a subsequent regenerative response. Is this implicated in the myogenic response observed by the authors to the administration of IL-10?

If the injection had a relevant contribution to the myogenic response, this would also be effective in the vehicle group, treated only with PBS.

In any case, the activation of SC myogenic response occurs spontaneously in ALS mouse models in the absence of mechanical injury. In the figure within THE PDF FILE ENCLOSED, we report that in the skeletal muscle of SOD1G93A mice, there is early activation of myogenic transcription factors, simultaneous to the generation of brand-new embryonic MyHC+ myofibres.

- The authors point out that they treat and extract three muscles (Tibialis Anterior (TA), Gastrocnemius Medialis (GCM), and Quadriceps (QC) (Page 3, line 130). However, they only refer to the TA muscle in the Immunohistochemistry procedures (page 3, lines (138-141), Western blotting (page 4, lines 185-187).

As spotlighted in the Results, Section 3.3:

"The TA was selected given the high composition in fast-fatigable fibres (IIb), which are early affected by the disease".

Based on Reviewer's notification, we have amended the sentence as follows in the Material and Methods, section 2.3:

"After being weighed TA, GCM and QC muscles, TA muscles only were used for immuno-histochemical, biochemical and biomolecular analyses."

-The authorisations for the study by the agencies are repeated (pages 2-3, lines 99-105) (page 3, lines 120-128).

We are sorry for the mistake. We remove the duplicate paragraph from Section 2.2 of the Materials and Methods.

-The abstract should be rewritten.

Based on Reviewer's criticisms, we have replaced in the title and the abstract the terms myogenesis and muscle regeneration as reported in the Reply above.

Round 2

Reviewer 1 Report

We thank the authors for the clarifications they included, as requested.

Reviewer 2 Report

The authors have been appropriately responsive to the comments. 

Reviewer 3 Report

I am satisfied with the revision and changes made by the authors to their manuscript.